# Inflammatory Markers in Uterine Lavage Fluids of Pregnant, Non-Pregnant, and Intrauterine Device Implanted Mares on Days 10 and 15 Post Ovulation

**DOI:** 10.3390/ani11123493

**Published:** 2021-12-08

**Authors:** Maria Montserrat Rivera del Alamo, Tiina Reilas, Karolina Lukasik, Antonio M. Galvão, Marc Yeste, Terttu Katila

**Affiliations:** 1Department of Animal Medicine and Surgery, Faculty of Veterinary Medicine, Autonomous University of Barcelona, 08193 Bellaterra, Spain; 2Natural Resources Institute Finland (Luke), 31600 Jokioinen, Finland; tiina.reilas@luke.fi; 3Institute of Animal Reproduction and Food Research, PAS, 10-748 Olsztyn, Poland; k.lukasik@pan.olsztyn.pl (K.L.); antonio.galvao@babraham.ac.uk (A.M.G.); 4Centre for Interdisciplinary Research in Animal Health, Faculty of Veterinary Medicine, University of Lisbon, 1300-477 Lisbon, Portugal; 5Epigenetics Programme, The Babraham Institute, Cambridge CB22 3AT, UK; 6Unit of Cell Biology, Department of Biology, Faculty of Sciences, University of Girona, 17003 Girona, Spain; marc.yeste@udg.edu; 7Department of Production Animal Medicine, Faculty of Veterinary Medicine, University of Helsinki, 59511 Saari, Finland; terttu.katila@helsinki.fi

**Keywords:** intrauterine device, inflammation, prostaglandins, cytokines, IL-10, inhibin A

## Abstract

**Simple Summary:**

While intrauterine devices (IUDs) are used to prevent disturbing oestrous behaviour in sport mares, their mechanism of action has not been elucidated. The presence of an embryo or an IUD prevents cyclooxygenase-2 (COX-2) and subsequently prostaglandin (PG) release and luteolysis. It has been suggested that a plastic sphere would mimic the embryo by mechanotransduction. However, there is some evidence that IUDs also cause endometrial inflammation, which might contribute to luteostasis. The aim of this study was to investigate the presence and time course of possible inflammation by evaluating changes in uterine fluid composition. On Day 10 after ovulation, events leading to COX-2 and prostaglandin F_2α_ (PGF_2α_) inhibition start, whereas either luteolysis occurs or the corpus luteum is maintained on Day 15. Therefore, uterine lavage fluid was evaluated at two time points in inseminated mares, either pregnant or not, and in mares inserted with an IUD. On Day 10, PGF_2α_ concentration in the fluid was significantly lower in the IUD group than in the pregnant mare one but did not differ from the non-pregnant mare group. On Day 15, the IUD group had significantly higher levels of the modulatory cytokine IL-10 and inhibin A, which could indicate previous inflammation and resolution stage.

**Abstract:**

Intrauterine devices (IUDs) are used in mares to suppress oestrous behaviour, but the underlying mechanism is yet to be elucidated. The presence of an embryo or an IUD prevents cyclooxygenase-2 (COX-2) and, subsequently, prostaglandin (PG) release and luteolysis. However, inflammation may also be involved. Endometrial inflammatory markers in uterine lavage fluid were measured on Day 10 (EXP 1, *n* = 25) and Day 15 (EXP 2, *n* = 27) after ovulation in inseminated mares, non-pregnant or pregnant, and in mares in which a small plastic sphere had been inserted into the uterus 4 (EXP 1) or 3 days (EXP 2) after ovulation. Uterine lavage fluid samples were analysed for nitric oxide (NO), prostaglandin E_2_ (PGE_2_) (only EXP 1), prostaglandin F_2α_ (PGF_2α_), inhibin A and cytokines, and blood samples for progesterone and oestradiol. On Day 10, the concentration of PGF_2α_ was lower (*p* < 0.05) in the IUD group than in pregnant mares. The concentration of the modulatory cytokine IL-10 was significantly higher in the IUD group in comparison to non-pregnant mares, and inhibin A was significantly higher in IUD mares than in the pregnant counterparts on Day 15. The results suggest that the presence of IUD causes endometrial inflammation which is at a resolution stage on Day 15.

## 1. Introduction

Intrauterine devices (IUDs) are used in sports mares to prevent unwanted oestrous behaviour [1]. Their efficacy, however, is highly variable; prolonged luteal phases have been reported in 0% to 75% of mares or cycles [1,2,3]. In a normal cycle, prostaglandin F_2α_ (PGF_2α_) is released from the endometrium as several spikes between Days 14 to 17 of the cycle resulting in luteolysis [4]. Both the embryo and IUD prevent PGF_2α_ release by inhibiting cyclooxygenase-2 (COX-2) [5,6] and, as a consequence, prevent luteolysis. Subsequently, the lifespan of the corpus luteum is prolonged, allowing the continuous production of progesterone and the maintenance of pregnancy or prolongation of dioestrus due to an IUD [2].

The mechanism behind equine maternal recognition of pregnancy (MRP) has not yet been elucidated, and, similarly, the way the IUDs work remains unresolved. Direct contact of the embryo with the endometrium, enhanced by actively moving around in the uterus, is probably important in the inhibition of luteolysis by attenuating the secretion of PGF_2α_ [7]. Interaction of the embryo with the endometrial wall involves mechanotransduction that induces changes in the abundance of endometrial focal adhesion molecules on Days 9 and 11 of pregnancy, decreasing PGF_2α_ secretion [8]. The plastic spheres also contact the endometrial walls, although they do not move much beyond the uterine body area [2]. Mechanotransduction has also been suggested as a possible mechanism for IUDs [2]. However, although plastic beads activated focal adhesion molecules, they did not alter PGF_2α_ secretion in an in vitro study [8].

Another explanation is chronic inflammation caused by the presence of an IUD [1,9], but the evidence for inflammation is not consistent. While Daels and Hughes [9] reported chronic endometritis after one-year use of the device in wild mares, some other studies [1,2,3] described the presence of intrauterine fluid but, after removal of IUDs, uterine lavage fluids [2] and biopsies were negative for leukocytes [1,3,10]. It is possible that the time point was too late to detect inflammation or that the methods used were not sensitive enough to diagnose chronic endometritis. A recent publication focused on the proteomic composition of the intrauterine fluid showed that IUDs induce annexin A1 secretion, an inflammatory mediator that contributes to the inhibition of luteolysis by acting as a phospholipase A2 inhibitor [11].

The aim of this study was to investigate the presence and time course of possible inflammation by evaluating changes in the uterine fluid composition. For this purpose, endometrial secretion was evaluated at two essential points of the cycle in inseminated mares as controls, either pregnant or not, and in mares inserted with an IUD. Around Day 10 after ovulation, events leading to COX-2 and PGF_2α_ inhibition start [12,13,14], whereas around Day 15 after ovulation, either luteolysis occurs or the corpus luteum is maintained [4]. Prostaglandins E_2_ and F_2α_, nitric oxide (NO) and both pro-inflammatory interleukins (IL-1α, IL-1ß and IL-8) and modulatory cytokine IL-10, as well as inhibin A, were measured at the two time points.

## 2. Materials and Methods

### 2.1. Animals

A total of 31 mares (Finnhorses and warmbloods) belonging to the Equine College and MTT Agrifood Research Finland (Ypäjä, Finland) were included in two experiments. Experiment 1 (EXP 1) included 25 mares and experiment 2 (EXP 2) 27. Thus, 21 mares underwent both experiments, while 4 and 6 were used only in EXP 1 and 2, respectively. Mares were from 4 to 17 years of age, clinically normal and with no history of reproductive failure. In both experiments, the mares were ranked according to their age, number of foals, and breed, then being alternately assigned into two groups: the intrauterine device group (IUD) (EXP 1: *n* = 12; EXP 2: *n* = 15) and the insemination group (AI) (EXP 1: *n* = 13; EXP 2: *n* = 12). The AI group was further divided into mares in which an embryo was found in the ultrasound and/or in uterine lavage (AI-P) and into the non-pregnant ones (AI-N) [6].

The study was approved by the Animal Ethics Committee of the Provincial Government of Southern Finland (number ESLH-2008-00122/Ym-23).

### 2.2. Experimental Design

#### 2.2.1. Experiment 1

The oestrus cycles of the mares were synchronised with one or two PG injections (cloprostenol, 0.125 mg, Estrumate vet^®^, Schering Plough A/S Farum, Denmark). Rectal palpation and ultrasonography (SonoSite Vet 180 plus; SonoSite Inc., Bothell, WA, USA) with a 5-MHz probe were started on Day 5 after PG administration and performed every other day. Once in early oestrus, examinations were performed daily. When a follicle of ≥35 mm in diameter was observed, a double-guarded uterine swab (Equi-Vet^®^, Kruuse, Marslev, Denmark) was obtained. The tip of the swab was then streaked onto a blood agar plate for microbiological purposes. After that, the swab was laterally rotated on a slide for cytological purposes. Mares were then examined daily until ovulation was detected. One day after swabbing, 1500 IU of hCG (Chorulon^®^, Intervet International B.V., Boxmeer, The Netherlands) were administered intravenously to time ovulation. The ovulation day was designated as Day 0. 

Mares in the AI group were inseminated approximately 24 h after hCG administration with fresh semen from a stallion of proven fertility. Mares in the IUD group received the intrauterine device on Day 4 after ovulation using the double-glove technique [15]. The IUD was a 20-mm polypropylene ball filled with sterile water (Euro-Matic UK, Ltd., London, United Kingdom) [2].

Blood samples from the jugular vein were collected on Days 0, 4, and 10. On Day 10, the inseminated mares were examined via ultrasound for the presence of an embryo, whereas IUD mares were examined to determine the location of the device. In addition, all the mares were examined for the presence of uterine fluid and oedema. Uterine lavages (30 mL PBS) were performed as described by Rivera del Alamo et al. [11], and embryos in pregnant mares were recovered. After the 30 mL lavage with PBS, the uterus of the AI mares was flushed one to three times with 1000 mL of Ringer’s acetate solution (Baxter Healthcare Ltd., Norfolk, UK) in order to recover the embryo in Experiment 1. 

Once uterine samples were obtained, a single i.m. injection of 0.125 mg of cloprostenol (Estrumat vet^®^, Schering Plough A/S, Farum, Denmark) was administered to induce luteolysis. During the induced oestrus, a uterine swab for microbiology and cytology was obtained, and the devices were removed from IUD mares by transrectal manipulation. Cytological smears were examined for the presence of polymorphonuclear leukocytes (PMN), and bacteria were cultured both according to Reilas and Katila [16].

#### 2.2.2. Experiment 2

Oestrus synchronisations, transrectal palpations and ultrasonography, and inseminations were performed as previously described in Experiment 1. The insertion of IUD was performed on Day 3. Blood samples from the jugular vein were obtained for progesterone and oestradiol analysis on Days 0, 3, 5, 7, 9, 11, 13, 14, and 15. On Day 14, ultrasonography was performed to establish the presence or absence of an embryo in the AI mares and the location of the device in the IUD mares.

On Day 15, a low-volume lavage was performed after transrectal palpation and ultrasonographical examination as in EXP 1.

### 2.3. Analyses in Lavage Fluid

Low-volume lavage fluids were kept on ice until centrifugation. A sample from the fluid was streaked onto a blood agar plate for bacteriological examination. One part of the fluid was centrifuged at 500× *g* for 10 min to prepare a cytological smear from the pellet. The remaining fluid was centrifuged at 1000× *g* for 20 min at 4 °C. The supernatant was dispended into EDTA, and heparin tubes and portions of the fluid were stored at −80 °C in Eppendorf tubes until analysis. 

ELISA was used to analyse the content of PGF_2α_, PGE_2_ (only in EXP 1), IL-1α, IL-1β, IL-8 (chemokine ligand 8), IL-10, and inhibin A in the uterine lavage fluid (Table 1). Nitric oxide was assessed using Griess Reagent System manufacturer’s instructions (G2930, Promega, Madison, WI, USA). The amount of NO produced was determined spectrophotometrically as formed nitrite/nitrate. The nitrite/nitrate content was calculated based on a standard curve constructed using NaNO_2_ (0–100 M). 

### 2.4. Analyses in Blood

Blood samples for progesterone and estradiol-17β were collected using 10 mL vacuum serum tubes (BD Vacutainer, BD-Plymouth, Plymouth, UK). Serum was separated by centrifugation at 2200× *g* for 10 min, divided into small portions, and frozen at −20 °C for further analyses.

Progesterone levels were determined with the Spectria Progesterone radioimmunoassay (RIA) kit (Orion Diagnostica, Espoo, Finland), using the 1270 Rackgamma counter (Wallac Oy, Turku, Finland). The detection limit was 0.7 nmol/L. The variation coefficients were 11.5%, 3.0%, and 3.8% (intra-assay) and 7.8%, 5.1%, and 4.8% (inter-assay) for low, medium, and high levels of progesterone, respectively.

Serum estradiol-17β concentration was determined with a commercially available human radioimmunoassay (Ultra-sensitive estradiol RIA, DSL4800, Immunotech a.s., Prague, Czech Republic) according to the assay procedure of the manufacturer. All samples were determined in duplicate in a single assay. The intra-assay coefficient of variation (CV%) was 8.7% and 6.1% calculated from six repeated measurements of two sera with different oestradiol concentrations (13.2 and 39.7 pg/mL, respectively).

### 2.5. Statistical Analysis

The results were evaluated using a statistical package (IBM SPSS for Windows Ver. 25.0; Armonk, NY, USA). Data were first tested for normal distribution (Shapiro–Wilk test) and homogeneity of variances (Levene test), prior t running one-way ANOVA followed by post-hoc Sidak test. In cases where the data did not fit with parametric assumptions, they were transformed with arcsin √x, and the corrected data were used to re-run the ANOVA and post-hoc Sidak test. In cases such as those of PGF_2α_ and IL-1α in EXP 1 and PGF_2α_ in EXP 2, in which linear transformation did not remedy normal distribution and homogeneity of variances, Kruskal–Wallis and Mann–Whitney tests were used as non-parametric alternatives of ANOVA and post-hoc Sidak test, respectively.

In all cases, the significance level was set at *p* ≤ 0.05.

## 3. Results

### 3.1. Experiment 1

Small numbers of bacteria were cultured from the lavage fluid of two mares (two and six colonies in one IUD and one AI-P mare, respectively), but they were not associated with endometritis since all the smears were negative for PMNs, and no uterine fluid or oedema was visible on the ultrasonography.

An embryo was found in 5 out of 13 inseminated mares, and five mares were diagnosed as non-pregnant. Three of the inseminated mares were excluded from the analysis of the results because it was not possible to accurately determine if they were pregnant or not.

Results for EXP 1 are shown in Table 2. No significant differences were observed among the three groups, with the exception of intrauterine levels of PGF_2α_, which showed significantly lower values for IUD mares than for AI-P mares.

### 3.2. Experiment 2

Results of EXP 2 are shown in Table 3. Mares with IUD showed significantly higher values of IL-10 than AI-N mares. The IUD group had significantly higher levels of inhibin A than the AI-P group. Levels of PGF_2α_ were significantly lower in IUD mares compared with AI-P mares. Progesterone and oestradiol concentrations from day 0 to 13 and from day 0 to 14, respectively, showed no significant difference among the different groups.

## 4. Discussion

In the present study, two time points were selected as sampling days: on Day 10, events leading to luteostasis start, whereas on Day 15, luteolysis commences. The equine embryo arrives at the uterus 6–6.5 days after ovulation [17], so on Day 10, it has influenced the uterus for four days. It is believed that the factor mediating MRP must exert its effect around Day 10 [18]. Systemic administration of oxytocin must be initiated before Day 10 to result in prolongation of the luteal phase [12]. Furthermore, intrauterine administration of peanut or coconut oil on Day 10 resulted in luteal persistence in 92% of treated mares. Vegetable oils have high contents of linoleic and capric acid, which are PG inhibitors [14]. Microarray analyses of endometrial biopsies and embryos suggest that the changes leading to MRP may slowly start between Days 8 and 10 [18,19]. Thus, the ability of the endometrium to react and prevent PGF_2α_ release has to be there already on Day 10. However, it has not been shown that IUDs follow the same timetable.

On Day 10, the intrauterine concentration of PGF_2α_ was significantly lower in mares carrying the IUD compared to pregnant mares, but although it was numerically lower than in non-pregnant mares, the difference was not significant, presumably because of the heterogenicity and small size of the non-pregnant group. Noteworthily, the IUD group consisted of mares that were destined to have corpus luteum (CL) maintenance and of those that would undergo luteolysis. Therefore, it is impossible to conclude if this finding suggests modulation of PG synthesis. Nonetheless, it suggests that inflammation was controlled on Day 10, or at least there was no acute inflammation [20,21].

Both the conceptus and endometrium of pregnant mares have been shown to produce PGF_2α_ as early as Day 10 onwards [13,22], but it remains in the uterine lumen. The ability of the endometrium of pregnant mares to produce PGF_2α_ in vitro is comparable or greater than that of the endometrium of non-pregnant mares, and thus a direct inhibitory effect of conceptus seems likely [22,23]. Since the only relevant difference among our groups was the presence of the embryo, it seems logical to assume that the high levels of PGF_2α_ on Day 10, and particularly on Day 15, were produced by the embryo itself.

In mares, PGE_2_ is luteotropic as an auto-paracrine factor stimulating progesterone production by luteal steroidogenic cells in vitro [24]. In the human bone tissue, PGE_2_ synthesis is increased in response to mechanical stimuli via the formation of local adhesions [25]. In the female reproductive tract, activation of a mechanosensitive channel, an epithelial sodium channel, triggers the release of PGE_2_ [26]. However, PGE_2_ levels—which were measured only in EXP 1—did not differ among the study groups. Either Day 10 was too early, or IUD did not affect PGE_2_ secretion.

Nitric oxide is involved in equine endometrial secretion modulating PGE_2_ and PGF_2α_ production [27]. It is also involved in the mechanism of persistent endometritis in mares, with susceptible mares showing higher levels in endometrial secretions 6 and 12 h after artificial insemination [21,28,29]. No differences in NO between inseminated, either pregnant or not, and device mares were observed. The most feasible explanation is the sampling time points. On Day 10, the IUD had been in the uterus for 6 days and on Day 15 for 12 days, so the early NO response to the device was no longer present if there was inflammation in the first place. On the other hand, NO synthase is inhibited by Annexin A1 (ANXA1) [30], which is increased in the intrauterine fluid of mares inserted with an IUD according to our previous results [11]. Thus, increased ANXA1 may inhibit NO synthesis, another explanation for the absence of differences in intrauterine levels of NO between inseminated and device mares.

Pro-inflammatory cytokines, including IL-1β, are secreted at the onset of post-breeding inflammation [21,31]. In the present study, differences among groups were observed neither on Day 10 nor on Day 15. This may indicate an absence of inflammation, but more likely, that their levels had already returned to the physiological status because they are secreted at the early phase of inflammation.

IL-8 or CCL8, also known as monocyte chemoattractant protein 2 (MCP2), is a chemoattractant cytokine for leukocytes [32]. Unlike IL-1, IL-8 is not under the control of IL-10 [33]. Mechanical stress and stretching induced secretion of IL8 from cultured endometrial stromal cells of women, which was counteracted by progesterone [34]. Thus, mechanical stretch induces biochemical signals in the endometrium leading to a pro-inflammatory environment [26,34]. No significant differences were detected between the inseminated and IUD groups, so IL-8 did not support acute inflammation or mechanotransduction.

In the present study, IUD mares showed a significantly higher abundance of IL-10 on Day 15 than non-pregnant mares but not on Day 10. The modulatory cytokine IL-10 inhibits inflammation by repressing the production of pro-inflammatory cytokines, such as IL-1, IL-6, IL-12, and tumour necrosis factor (TNF) [35]. Previous studies have shown that endometrial expression of IL-10 increases in experimentally induced endometritis in mares, contributing to the decrease of the expression of pro-inflammatory cytokines [36]. Likewise, IL-10 has also been suggested to play a role in spontaneous chronic endometritis in mares [37]. The absence of pro-inflammatory cytokines on Day 10 (6 days after IUD insertion) shows that the acute phase was over. On the other hand, inhibitory cytokines should have been found. The inhibitory stage may have also been over so many days after the initial insult, but the higher levels of IL-10 in the IUD group on Day 15 contradict this assumption.

IL-10 release is stimulated by Annexin A1 (ANXA1) (reviewed by [38]), which, as mentioned above, shows higher intrauterine concentrations in mares carrying an intrauterine device [11]. In addition, to stimulating IL-10 release, ANXA1 is involved in the resolution of inflammation by means of limiting the recruitment of neutrophils and the production of pro-inflammatory factors, among other mechanisms (reviewed by [39]). On the other hand, because annexin A1 is a phospholipase inhibitor, it can prevent COX-2 and subsequent PGF_2α_ release [39]. By this mechanism, inflammation could be luteostatic.

Other findings reinforcing the hypothesis of chronic endometritis are those reported by Klein et al. [10], who observed a significantly higher endometrial expression of uteroferrin in mares implanted with an IUD. 

Thus, previous results of higher levels of ANXA1 plus higher levels of IL-10 in endometrial secretions from IUD mares suggest that IUDs cause inflammation. However, it cannot be determined whether IUDs cause acute inflammation that evolves to a chronic status or if the inflammation starts as chronic since samples were not obtained during the time frame when the acute stage occurs.

Inhibin belongs to the transforming-growth-factor-α (TGF-α) family. Its expression has been described in many organs, including ovaries and the uterus of many animals, among others in the mare [40]. In women, inhibin A antagonises the effect of activin on metalloproteinases (MMP) in addition to directly inhibiting MMP-2 [41]. MMPs are involved in the extracellular matrix remodelling of the endometrium [42]. In the present study, IUD mares showed higher levels of endometrial inhibin A than pregnant mares. Inhibin A has been suggested as a potential marker for endometritis in post-partum cows, but it also shows ongoing active remodelling and restoration of endometrium after parturition and inflammation [43]. One could thus speculate that inhibin A indicates previous inflammation and the need for endometrial restoration in IUD mares. 

Thus, in EXP 1, there was no evidence of inflammation. In EXP 2, pro-inflammatory cytokines were not elevated, but IL-10 and inhibin A were. The IUD was present in this experiment 6 days longer than in EXP 1 and may have induced a more chronic or severe inflammation. 

## 5. Conclusions

The low levels of NO, PGF_2α,_ IL-1, and IL-8 speak for the absence of acute inflammation. However, on Days 10 and 15, the IUD had been in the uterus for six and twelve days, respectively, and therefore, the possible inflammation would be of chronic nature at that time. There was no evidence for inflammation on Day 10, but elevated IL-10 and inhibin A levels on Day 15 could indicate inflammation at the resolution stage. No clear evidence for mechanotransduction or modulation of PG secretion by IUD was detected, although the antiluteolytic action of IUDs in those mares in which IUD is effective has been shown earlier. In this study, the IUD group included both mares that responded to the IUD by the maintenance of the CL and mares in which IUD did not prevent PGF_2α,_ release. Therefore, the combined results of this group did not show lower PGF_2α,_ values than the non-pregnant group. It is very likely that the presence of IUDs leads to chronic inflammation, which is at a resolution stage at the time of expected luteolysis. This may contribute to the inhibition of luteolysis, although further research is warranted to completely elucidate the actual implication of chronic inflammation as the luteostatic mechanism behind the IUDs.

## Figures and Tables

**Table 1 animals-11-03493-t001:** RIA/ELISA kit names for analyses. CVs: intra and inter-assay coefficients of variation; E_2_: estradiol-17α; NO: nitric oxide; PG: prostaglandin; IL: interleukin.

	RIA/ELISA Kit Name	Manufacturer Product No	Curve Range	CVs (Inter-/Intra-)
E_2_ (pg/mL)	Ultra-sensitive estradiol RIA, human	Immunotech a.s.; DSL4800	0.5–750	8.7%/6.1%
PGF_2α_ (pg/mL)	Prostaglandin F_2α_ (PGF_2α_) ELISA kit	Enzo; ADI-901-069	3.05–50,000	13.1%/9.7%
PGE_2_ (ng/mL)	Highly sensitive Prostaglandin E_2_ (PGE_2_) ELISA kit	Enzo; ADI-900-001	31.9–2500	17.5%/5.1%
IL-1α (pg/mL)	ELISA Kit for Interleukin 1 Alpha (IL1α), Equine	Cloud-clone; SEA071Eq	15.6–1000	10.0%/12.0%
IL-1β (pg/mL)	ELISA Kit for Interleukin 1 Beta (IL1β), Equine	Cloud-clone; SEA563Eq	15.6–1000	10.0%/12.0%
IL-8 (pg/mL)	Horse C-C Motif Chemokine 8/MCP2 (CCL8) ELISA Kit, Equine	Abbexa; abx575576	15.6–1000	10.0%/12.0%
InhibinA (pg/mL)	Inhibin A (Equine, Canine, Rodent) ELISA	AnshLabs; AL-161	6.6–668	10.0%/12.0%

**Table 2 animals-11-03493-t002:** Mean (±SEM) values of the parameters on Day 10 in Experiment 1. Statistical differences are marked with different superscripts. All other analyses were performed in uterine lavage fluid. AI-N: inseminated non-pregnant mares; AI-P: inseminated pregnant mares; IUD: device mares; S P_4_ D_4_: serum progesterone on Day 4 after ovulation; S P_4_ D_10_: serum progesterone on Day 10 after ovulation; S E_2_ D_10_: serum estradiol-17β on Day 10; NO: nitric oxide; PG: prostaglandin; PGE/PGF: ratio of PGE and PGF_2α_; IL: interleukin. Statistical differences are marked with different superscripts.

Parameter	AI-N *n* = 5	AI-P *n* = 5	IUD *n* = 12
S P_4_ D_4_ (nmol/L)	23.2 ± 2.82	19.6 ± 2.95	21.7 ± 1.96
S P_4_ D_10_ (nmol/L)	18.3 ± 3.06	17.4 ± 1.40	16.6 ± 1.53
S E_2_ D_10_ (pg/mL)	18.57 ± 2.75	19.35 ± 1.88	16.57 ± 1.52
NO (ng/mL)	5.47 ± 1.30	7.15 ± 1.46	6.99 ± 1.45
PGF_2α_ (ng/mL)	0.85 ± 0.22 ^ab^	0.95 ± 0.03 ^a^	0.45 ± 0.09 ^b^
PGE_2_ (ng/mL)	122.40 ± 22.96	130.46 ± 10.50	119.76 ± 12.43
PGE/PGF IL-1α (pg/mL)	192.07 ± 77.92 54.69 ± 21.45	143.80 ± 8.76 8.22 ± 3.33	289.91 ± 47.11 14.47 ± 8.69
IL-1β (pg/mL)	27.77 ± 4.11	23.02 ± 3.33	19.17 ± 2.90
IL-8 (pg/mL)	10.60 ± 3.11	5.90 ± 2.91	7.31 ± 3.18
IL-10 (pg/mL)	14.71 ± 2.92	10.33 ± 0.52	10.62 ± 1.38
Inhibin A (pg/mL)	11.47 ± 1.32	14.31 ± 2.35	17.59 ± 2.00

**Table 3 animals-11-03493-t003:** Mean (±SEM) values of the parameters on Day 15 in Experiment 2. Statistical differences are marked with different superscripts. S: serum, all other analyses were performed in the intrauterine fluid. AI-N: inseminated non-pregnant mares; AI-P: inseminated pregnant mares; IUD: device mares; P_4_ D_14_: progesterone on Day 14; P_4_ D_15_: progesterone on Day 15; E_2_ D_15_: estradiol-17β on Day 15; NO: nitric oxide; PG: prostaglandin; IL: interleukin. Statistical differences are marked with different superscripts.

Parameter	AI-N *n* = 4	AI-P *n* = 8	IUD *n* = 15
S P_4_ D_14_ (nmol/L)	7.25 ± 2.32	14.88 ± 2.08	13.57 ± 2.03
S P_4_ D_15_ (nmol/L)	6.75 ± 3.15	16.0 ± 2.79	9.07 ± 2.26
S E_2_ D_15_ (pg/mL)	23.42 ± 3.58	19.90 ± 2.24	15.65 ± 1.34
NO (ng/mL)	7.70 ± 2.06	7.92 ± 2.83	5.10 ± 0.98
PGF_2__α_ (ng/mL)	1.72 ± 0.32 ^a^	16.70 ± 4.69 ^b^	1.81 ± 0.49 ^a^
IL-1α (pg/mL)	33.16 ± 19.07	42.07 ± 22.33	14.22 ± 6.36
IL-1β (pg/mL)	14.72 ± 3.58	18.99 ± 2.94	19.24 ± 1.55
IL-8 (pg/mL)	7.44 ± 3.76	7.33 ± 2.25	13.32 ± 2.92
IL-10 (pg/mL)	7.41 ± 1.00 ^a^	9.47 ± 0.82 ^ab^	10.51 ± 1.81 ^b^
Inhibin A (pg/mL)	19.90 ± 5.36 ^ab^	12.59 ± 3.33 ^a^	25.02 ± 2.93 ^b^

## Data Availability

Data are available under request to the authors.

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
