# Peer review of "Inflammatory Markers in Uterine Lavage Fluids of Pregnant, Non-Pregnant, and Intrauterine Device Implanted Mares on Days 10 and 15 Post Ovulation"

_animals, 2021, doi:10.3390/ani11123493_

Round 1
Reviewer 1 Report
Dear Drs Rivera del Alamo and colleagues,
General comments
This is a very nice study, well planned, executed and presented which reveals some interesting data to drive further research in this area.
Specific comments
In the methodology, under section 2.3 Analyses in lavage fluid, lines 161-162: 'After the 30-mL lavage with PBS, the uterus of the AI mares was flushed one to three times with 1000 mL of Ringer’s acetate solution (Baxter Healthcare Ltd, Norfolk, UK).' I am assuming that this was to recover the embryo(s) and the analyses were performed on the 30ml PBS and not the recovered Ringer's acetate solution? Perhaps the above detail would be better placed in the description of Experiment 1 lines 135-136?
In the description of Experiment 2, you describe the taking of blood for progesterone and oestradiol lines 147-148: 'Blood samples from the jugular vein were obtained for progesterone and oestradiol analysis on Days 0, 3, 5, 7, 9, 11, 13, 14, and 15.' The results of these analyses are not presented for days 0-13 for P4 and days 0-14 for E2 respectively. One assumes there were no significant differences in levels hence no further comments provided?
Author Response
Dear Drs Rivera del Alamo and colleagues,
General comments
This is a very nice study, well planned, executed and presented which reveals some interesting data to drive further research in this area.
Answer: Dear reviewer, thanks for your kind comment and we hope all your concerns are solved with the following answer.
Specific comments
In the methodology, under section 2.3 Analyses in lavage fluid, lines 161-162: 'After the 30-mL lavage with PBS, the uterus of the AI mares was flushed one to three times with 1000 mL of Ringer’s acetate solution (Baxter Healthcare Ltd, Norfolk, UK).' I am assuming that this was to recover the embryo(s) and the analyses were performed on the 30ml PBS and not the recovered Ringer's acetate solution? Perhaps the above detail would be better placed in the description of Experiment 1 lines 135-136?
Answer: You are right: flushing with Ringer’s acetate was performed to recover the embryos. The text has been modified accordingly and we hope it is clearer now. The sentence on line 161 has been modified in order to clarify this.
In the description of Experiment 2, you describe the taking of blood for progesterone and oestradiol lines 147-148: 'Blood samples from the jugular vein were obtained for progesterone and oestradiol analysis on Days 0, 3, 5, 7, 9, 11, 13, 14, and 15.' The results of these analyses are not presented for days 0-13 for P4 and days 0-14 for E2 respectively. One assumes there were no significant differences in levels hence no further comments provided?
Answer: Correct. No differences were observed during these days and consequently they were not included in the text. It has been specified in the text
Reviewer 2 Report
Dear authors,
thank you for the interesting article "Inflammation induced by intrauterine devices in mares is at resolution stage on Day 15 post-ovulation" submitted to Animals. The paper addresses an important topic and is well-written. However, I have got some minor remarks and questions:
General questions/remarks:
- While performing the experiment, did you perform gynecological examinations including ultrasound for detection of signs of endometritis (e.g. i.u. fluid, hyperoedema)?
- The title of the manuscript is only partly supported by your results in my opinion. Please consider a more precise title. To me, it is highly speculative that mares are in resolution stage 5 days after the first sampling date, when no signs of endometritis were noted in the IUD group.
- In the M&M section, you reported the collection of cytology samples, that did not appear in your results section. Please add the cytology results to the results section and include them into discussion.
specific remarks:
l. 67: citation as number misses.
l. 122-124: please explain your swabbing technique.
l. 125: please add the application route of hCG.
l. 129: Did you use native / fresh / chilled / frozen thawed semen?
l. 131: Please indicate the manufacturer.
l. 146-147: Why did you insert the IUDs one day earlier in experiment 2? Does this possibly interfere with your results?
l. 171: Please indicate the manufacturer / type of collection tube.
Author Response
Dear authors,
thank you for the interesting article "Inflammation induced by intrauterine devices in mares is at resolution stage on Day 15 post-ovulation" submitted to Animals. The paper addresses an important topic and is well-written. However, I have got some minor remarks and questions:
Answer: Dear reviewer, thanks for your kind comment and we hope all your concerns are solved with the following answer.
General questions/remarks:
- While performing the experiment, did you perform gynecological examinations including ultrasound for detection of signs of endometritis (e.g. i.u. fluid, hyperoedema)?
Answer: Yes, on the ultrasound examination, the presence of uterine fluid, oedema, etc, were also evaluated. However, none of the mares showed any clinical sign of endometritis. This has been also clarified in the chapter of results.
The title of the manuscript is only partly supported by your results in my opinion. Please consider a more precise title. To me, it is highly speculative that mares are in resolution stage 5 days after the first sampling date, when no signs of endometritis were noted in the IUD group.
Answer: following your advice, the title has been modified.
- In the M&M section, you reported the collection of cytology samples, that did not appear in your results section. Please add the cytology results to the results section and include them into discussion.
Answer: The sentence in the results chapter has been modified to clarify that point
specific remarks:
- 67: citation as number misses.
Answer: citation number has been added.
- 122-124: please explain your swabbing technique.
Answer: The swabbing techniques has been explained in the text.
- 125: please add the application route of hCG.
Answer: this information has been added to the text.
- 129: Did you use native / fresh / chilled / frozen thawed semen?
Answer: it was fresh semen. This information has been added to the text.
- 131: Please indicate the manufacturer.
Answer: This information has been added to the text
- 146-147: Why did you insert the IUDs one day earlier in experiment 2? Does this possibly interfere with your results?
Answer: In our previous study (Rivera del Alamo et al., 2008), the device was inserted between day 2 and 4 after ovulation detection and no significant difference was observed. On the other hand, we changed the insertion day to Day 3 because the cervixes of some mares in EXP 1 were quite tight on Day 4.
- 171: Please indicate the manufacturer / type of collection tube.
Answer: this information has been added to the text
Round 2
Reviewer 2 Report
Dear authors,
thanks for the submission of the revised article and changing the title of the manuscript.